# The Circularity of MSW in Urban Landscapes: An Evaluation Method for a Sustainable System Implementation

Elvira Nicolini

Department of Architecture, University of Palermo, 90128 Palermo, Italy; elvira.nicolini@unipa.it

**Abstract:** The debate on the sustainable perspective of a circular economy and MSW (Municipal Solid Waste) management is no longer focused on the resolution of an emergency and on the pure technicality of creating efficient services. New light has been shed on the operational phase, as often unresolvable technical-environmental issues during service management lead to a state of decay, affecting urban vulnerability and users' involvement. The research aims to develop an auxiliary tool for public administrations in planning a zero-waste future whilst enhancing the urban landscape. This study considers the perspective of virtuous management of the resource flow cycle along with evaluating implementable theoretic models, operational scenarios, and technological tools that can be integrated into a specific built environment. The first phase consisted in the definition of the involved subjects and the evaluation criteria for the impact of the service management system. The second phase includes the definition of operational strategies for virtuous and sustainable management of resource flows, intended as opportunities for the regeneration and development of the local context. The result of this research work is the elaboration and experimentation of an information system for the evaluation of the integrability of operational scenarios with the systemic, quantitative, and qualitative elements that characterize an urban landscape. The selected actions and technologies have been assessed in terms of short-term feasibility, cost-effectiveness, and local efficacy. This research work shows that many urban contexts with efficient services have been conceived as ecosystems, integrated into the landscape, driven by the synergy of institutions and operators, capable of carrying out the entire cycle from waste production to the marketing of recycled products. Our analysis shows that this issue can be effectively tackled through varied and mixed solutions; by integrating strategies, techniques, and scenarios, and by stimulating citizens' involvement.

**Keywords:** MSW; circular metabolism; multi-criteria approach; resource flows; AHP

## 1. Introduction

Resource flow management is the set of actions aimed at managing the whole cycle of municipal solid waste (MSW): from the production phase to the collection, treatment, and landfilling or recycling phase. A virtuous, circular management aims to cancel—or at least limit—the final residual of this cycle and the related environmental and sanitary impacts throughout all the phases, reducing waste production and recovering resources from it. Virtuous management includes production prevention dynamics oriented to the citizens, promoting recycling and reuse of consumed products [1], involving industries, and banning the market introduction of products with a short life cycle. Several socio-economical evaluations have demonstrated that, if the end of the cycle is the transformation of the waste into a resource, there are improved effects on waste prevention, compared to landfilling or energy recovery. Waste sorting involves consumers in the management chain and induces them to reduce waste production [2]. However, the involvement of heterogeneous subjects can lead to organizational or planning errors, in addition to market problems that can affect the urban economy and contribute positively to the national economy [3].

The economy is determined by continuous material flows: some are extracted inside the country while others are imported; waste is the result of the production and disposal operations of these flows. In the EU, the overall production of urban waste has increased throughout the years: 1.3% since 2018, from 221.6 million tons to about 224.4; compared to 2017, the increase is 1.6%. The average value of per capita urban resource flow production is around 500 kg/person: out of these, 119 kg/person are landfilled [4]. The overall framework is variegated, and few member States have an efficient management system with recycled quantities of 80%; however, in others up to 90% is landfilled. In Italy, the magnitude of the solid material flows throughout production and disposal is around 29.6 million tons [5]. The overall situation is inhomogeneous and lagging: this is unsustainable considering the current environmental hazard. Waste must be considered as an opportunity rather than a problem, leading to reorganizing the metabolism of urban areas in a circular process, optimizing resources, and minimizing residuals.

In urban centers, the issue of the cycle closure of material flows has radically increased over the last decades, following a progressive growth of housing density and technological and industrial waste production, which can be hardly disposed of [6]. In addition to being a cause of environmental decay, it often produces landscape alterations and depression in cultural, economic, and social relationships within communities [7]. Likewise, the management of this problem is often strictly related to the characteristics of the single territories, with different conformations and infrastructural, technological, organizational backgrounds. Sustainable development imposes a new perspective toward the future, involving the critical reconstruction of the fragmentary aspects of contemporaneity, yet keeping sedimented cultural and material identities [8]. The morphological characteristics of a city represent a set of sedimented values, rooted over the centuries, where the community recognizes and identifies itself. The circular reorganization of a MSW service can represent an opportunity for the regeneration of a landscape: in this way, it is involved in circularity itself, turns into a resource, an attractor, and an active part of the local economic development [1,9]. This opportunity must be valorized through long-term planning, in which tangible (technologies and infrastructures) and intangible (socio-cultural identities and relational factors) components are exalted and combined [10].

The goal outlined in this research work is the development of a systemic model to evaluate the integrability of a material flow management network in an historically dense and fine-grained urban core. In particular, this paper describes the experimentation of a method that allows preliminarily comparing the impacts of different strategies and represents a decision support system. Different scenarios of waste disposal and energy recovery are here hypothesized, considering both the cycle closure of recyclable materials (zero waste objective) and local resource recycling (principle of proximity). The experimentation evaluates the integration into an urban context for each scenario and the optimization of the logistic route of waste. The evaluation is preceded by the definition of organized and comparable criteria, aimed at the protection and enhancement of the urban landscape. The case study is centered on the historical center of Palermo, in Sicily, Italy, and included the development of a specific analysis regarding the resource flow management at an urban level (uses, services, users, sorted waste categories, and quantitative estimates) and the morphological and infrastructural characteristics of the spaces of the housing fabric. The simulation of collection and pre-treatment flows with systems based on the combination of different technologies led to planning a network infrastructure for the internal connection of the urban center and its external connection with the rest of the city, concerning waste management. The major criticality has been the integration of technological systems in context with many physical, environmental, and perceptive-cultural restrictions.

## 2. Method

The paper proposes a method for the prior evaluation of operational models for the management of the waste collection service, aiming at the valorization of an urban landscape through conscious choices.

The resilience of an urban system is proportional to the entity of the conservation of the intrinsic cultural capital [11], also to the economic dimensions too. Service planning requires considering various dimensions (morphological, environmental, and social) that define the characteristics of the city and its inhabitants' needs. This required the implementation of multi-criteria analysis, which is a tool for the evaluation of alternative scenarios by considering the multiple dimensions of an urban landscape according to various decisional criteria.

In an urban context, conservation is performed on sets of buildings and open spaces, which are parts of urban settlements. The protected object is hence a morphological, functional, and typological asset, whose elements are non-substitutable parts of an organic whole, that is the city. Climate change and the fulfillment of hygienic-sanitary conditions require rethinking system logic: this is also more relevant in densely urbanized public spaces. Environmental incident criteria evaluate the compatibility of the characteristics of a given intervention with the standard physiological needs of environmental wellness, and with the conditions of full collective fruition of places. Criteria related to the social dimension evaluate possible human discomfort related to negative management of the MSW collection service. In addition to the definition of the general goal, the first phase includes the identification of the evaluation criteria for the estimation of the landscape impact of the single operational scenarios to compare them [12] (see Figure 1).

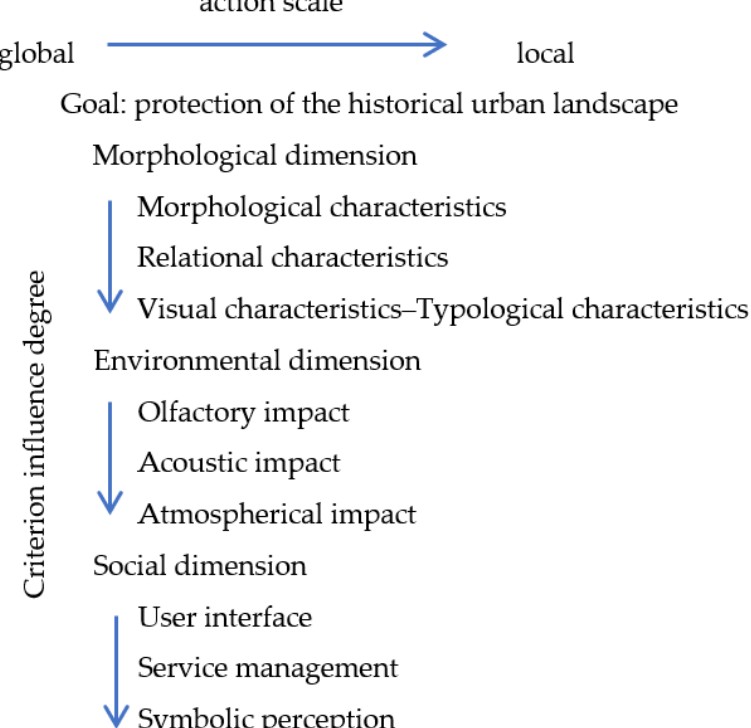

**Figure 1.** Hierarchy of the criteria. The fixed elements are ordered in a tree or pyramidal hierarchy according to their degree of abstraction: hence, the top items are abstract and general, while the bottom ones are tangible and definite and evaluate the relationship between the planned hypothesis and its context. The various criteria, organized in various levels, have a dependency property: that is, each level depends on its superior level; elements at the same level are interdependent. The impact of waste on human physiological health is transversal on all dimensions.

A new organization of the waste management service can sensibly impact both the environment and its socio-economic dimension [13], benefitting local development and leading to the creation of new economies. The related incomes for the collectivity could increase along with the practice of waste sorting. The realization of this scenario requires the direct involvement of local communities in the management chain of the cycles by local authorities, starting from the prevention of material consumption. The choice of the

evaluation criteria must consider needs and performances: that is, users' needs and their fulfillment by the performances of the service within the planned organization. For the sake of clarity, the paper defines the subjects involved in the material flow management process and describes their role.

The second phase consists in the configuration of the alternative scenarios, by individuating the possible collection infrastructures and the pre-treatment of the disposed materials, simulating circular material flows for each system. The scenarios are compared by hierarchizing criteria according to a pairwise comparison mathematical model, whose output is a symmetric matrix. This matrix is defined through the Analytic Hierarchy Process (AHP) [14], which determines an order of importance ranking among criteria by assigning a value of relative preferability to each of them. In this analysis, the preferability of one criterion to another is related to the goal of protection of the urban landscape.

AHP is a multicriteria analysis method that allows hierarchizing decision alternatives, relating the criteria characterized by multi-dimensional qualitative and quantitative variables, which cannot be directly compared [15].

The analysis process decomposes the problem into multiple elements, then recomposes them in a pyramidal hierarchy: at its vertex is the general goal, while the criteria are in the levels below.

The qualitative levels of Saaty's fundamental scale range from 1 to 9: they correspond to the (mathematical) differences between the values attributed to the single scenarios for each criterion. Specifically, the pairwise comparisons between scenarios lead to the definition of the differences between the scores defined during the single-scenario evaluation phase. For the sake of simplicity, the fundamental scale for pairwise comparisons is reduced from 9 to 5 levels. Given $i$ and $j$ compared scenarios, $Ci$ and $Cj$ their criteria and the reduction of landscape impact as the general goal for the comparison, the importance of each criterion toward the superior hierarchic level is defined as follows:

- value 1: $Ci$ and $Cj$ have equal importance;
- value 3: $Ci$ has minimally higher importance than $Cj$;
- value 5: $Ci$ has moderately higher importance than $Cj$;
- value 7: $Ci$ has strongly higher importance than $Cj$;
- value 9: $Ci$ has extremely higher importance than $Cj$;

Pairwise comparisons are based on determining which criterion has better properties between the two scenarios and hence fulfills better the goal. Given C1, C2, . . . , C$n$ the criteria of a given dimension and a given scenario, with corresponding $w_1$, $w_2$ . . . . . . $w_n$ values, the pairwise comparisons between the C$n$ elements lead to the construction of the $A_{ij}$ square matrix, whose elements are $a_{ij} = w_i/w_j$, which express the dominance of the $Ci$ element over $Cj$. This dominance is expressed through the fundamental scale [14].

The association of one of the 5 values of the scale to the criteria is related to a qualitative judgment. A quantitative judgment (from 1 to 5 as well) has then been determined on its base, through descriptors (see Table 1).

**Table 1.** Descriptors of urban landscape impact.

| Class | Morphological Impact Descriptor | Value |
|---|---|---|
| I. Very low impact | The criterion is scarcely applicable because the intervention protects specificity and coherently participates in the organization of the historical-cultural elements that characterize the morphology of the urban landscape. | 1 |
| II. Low impact | The criterion cannot be clearly identified, or it is blended among landscape elements; it provides an indifferent contribution to the characterization of the urban system. | 2 |

**Table 1.** *Cont.*

| Class | Morphological Impact Descriptor | Value |
|---|---|---|
| III. Medium impact | The criterion is visible, and its presence could negatively affect the protection of the identification and recognition of the landscape quality of the urban spatial system. | 3 |
| IV. High impact | The criterion is recognizable and its presence disturbs the continuity of the morphological, formal, and constitutive characteristics of the urban space. | 4 |
| V. Very high impact | The criterion is clearly recognizable, and its presence produces a significant alteration of the elements and the structural relationships of the place and territorial context. | 5 |
| **Class** | **Environmental impact descriptor** | **Value** |
| I. Very low impact | The criterion is scarcely recognizable, and its presence slightly affects the perceptive relationships between the site and its observers. | 1 |
| II. Low impact | The is partially visible; some potential alterations of environmental levels could hinder the total fruition of part of the urban space. | 2 |
| III. Medium impact | The criterion can be perceived and could lead to exceeding the minimum environmental levels of the site or part of it, negatively affecting landscape quality. | 3 |
| IV. High impact | The criterion is recognizable and its presence leads to exceeding regulatory limits, hindering the fruition of the urban space. | 4 |
| V. Very high impact | The criterion can be clearly affirmed and its presence impedes the overall sensorial fruition of the landscape-environmental context of the urban space. | 5 |
| **Class** | **Social impact descriptor** | **Value** |
| I. Very low impact | The criterion is scarcely affirmable and the intervention is considered to be coherent with lo spirit of the place, as it allows users to employ it with awareness and perception of the local identity. | 1 |
| II. Low impact | The criterion is partially affirmable, and its presence interferes with the symbolic value that local and supralocal communities might attribute to the place; however, the fruition of the site and services are guaranteed. | 2 |
| III. Medium impact | The criterion is perceivable and its presence limits service management, as it decreases the user-perceived wellbeing and landscape quality. | 3 |
| IV. High impact | The criterion is recognizable and it impedes the correct fruition of services, negatively affecting social well-being and the capacity of the place to recall expressive values associated with it by the local community. | 4 |
| V. Very high impact | The criterion is considerably present and leads to the loss of the symbolic and celebrative values of the place and the territorial context. | 5 |

Descriptors allow classifying the impact degree according to five levels, whose increasing order is related to a negative impact on landscape quality protection. The Table 1 reports the outline of the five impact levels for morphological, environmental, and social restrictions for the quantitative evaluation of the criteria.

The values of the criteria are not calculated through a direct measurement system, but rather on the base of qualitative judgments on the $a_{ij}$ element on the fundamental scale. The real weight $w_n$ can be calculated from the judgments through the expression:

$$w n = \frac{\sum_j a_{ij} w_j}{\lambda_{max}}$$

where $\lambda_{max}$ is the maximum eigenvalue of the $A_{ij}$ matrix. The maximum eigenvalue can also be used to verify the coherency of the matrix, that is its reliability. However, pairwise comparisons could result in incoherent judgments, in contradiction with the proportionality of the elements. This is due to the limits of the human mind, which cannot

consider all the relationships between elements at the same time [16]. The identification of the incoherency degree is followed by setting its tolerance. Coherency deviation is displayed by the coherency index: $CI = (\lambda_{max} - n)/(n - 1)$.

For each value of the fundamental scale (VFS) (here reduced to 5 scores, see Table 2), the calculator of the AHP process defines a random index RI, by calculating the average of the CI values of several randomly generated reciprocal matrices of the same order.

**Table 2.** Value of the fundamental scale.

| VFS | 1 | 3 | 5 | 7 | 9 |
|-----|---|---|---|---|---|
| RI  | 0 | 0.58 | 1.12 | 1.32 | 1.45 |

The ratio between CI and RI provides the coherency ratio RC of the matrix, which must be lower than 0.10 to be acceptable. If this threshold is exceeded, the pairwise comparisons must be reformulated, leading to the elaboration of a new matrix. The evaluation is concluded when an acceptable tolerance value is reached, resulting in a hierarchical scale of the hypothesized scenarios.

## 3. Discussion

### 3.1. Evaluation Criteria for the Defense of the Urban Landscape

The evaluation criteria for the comparison between alternative scenarios with respect to the goal of protection of urban landscape are detailed as follows. The protection of the morphological condition is evaluated in the hypothesized scenarios by assessing the coherency or contrast of the project with the physical rules of the place. This is expressed by the conservation or compromission of the fundamental and recognizable elements of the territorial morphological systems, in terms of visual burden, typological, chromatic, stylistic assonance or dissonance, etc. (see Figure 2).

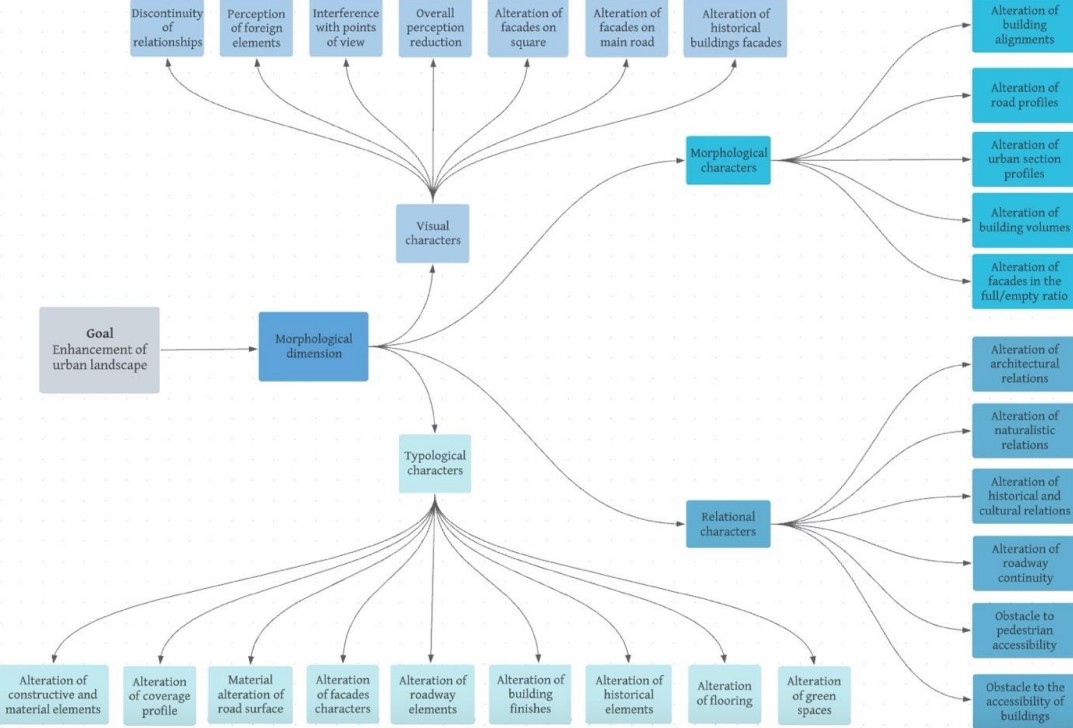

**Figure 2.** Criteria related to the morphological dimension for the evaluation of the impact of the resource flow management service on the urban landscape. The chosen criteria are aimed at protecting the morphological aspects of the urban landscape. At the vertex of the hierarchy is the general goal of the decisional process. (Source: Author's processing).

The compromission might occur in cases of negative management of the municipal solid waste collection service or the use of devices that impacting the landscape, for example. The evaluation considers both the internal coherency of the morphological and typological structure of the project with the system of the urban space and the external coherency, that is its relationship with the wider urban context. The internal coherency check reveals small-scale incongruencies, whose summation can produce a critical distortion of the overall vision of characteristic elements and their mutual relationships. In the external coherency check, the impact is evaluated in terms of reduction or obstacle to panoramic perception by extraneous elements to the urban historical landscape. The other examined elements include the possible interference with significant observation points on historically and culturally relevant public and private spaces, which allow assessing the integration of the new infrastructure in the context.

Urban resource flow collection systems can differ from each other by the modality of use of devices and vehicles, by the times and settings of service management: this can affect environmental quality in various ways, through the emission of pollutants in the atmosphere and noise production. Criteria related to the protection of the environmental restriction are aimed at individuating negative effects of the systems during the fruition of the urban historical landscape.

Negative environmental impacts (see Figure 3a) can arise in two different phases. The first one is related to the implementation of the technologies required for the operation of the service and/or the realization of a system network: this phase would have a temporary nature but would require the installation of a temporary construction site, with strong consequences on acoustic, olfactory, and atmospheric comfort. The second one is associated with the modalities of service management and the direct and indirect variables that could affect environmental quality during collection and pre-treatment—if present—phases.

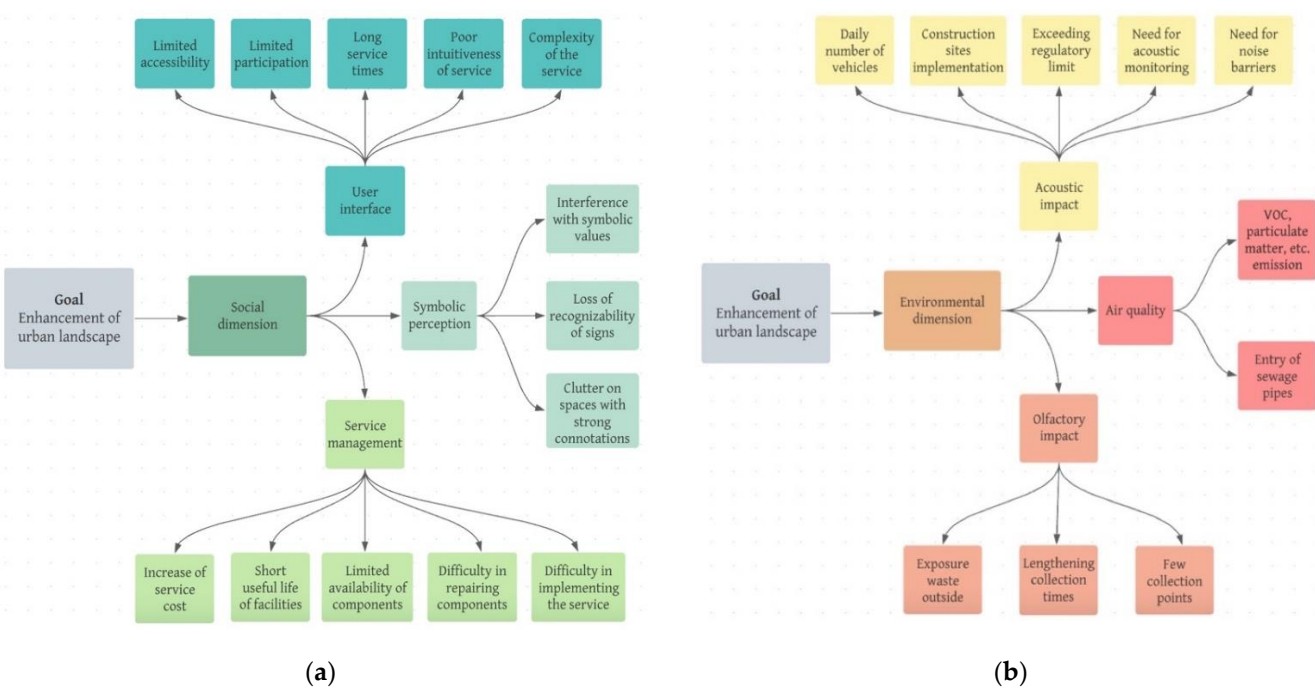

(**a**) (**b**)

**Figure 3.** Evaluation criteria for the impact of the resource flow management service in the urban landscape. (**a**) Criteria for the environmental dimension of the urban landscape. (**b**) Criteria for the social dimension of the urban landscape. Source: Author's processing.

For example, a possible cause of acoustic discomfort could be caused by an unfavorable variance of the daily number of vehicles per km of road network or number of inhabitants, or by temporary activities required or the realization or maintenance of the system.

Acoustic pollution monitoring or noise protection systems would be required. There could be other kinds of interferences: for example, an olfactory interference intended as the sensible form of aerial pollution, caused by the exposition of fermented material to the external environment, in particular when an extension of collection time or an increase in the number of waste containers and/or waste separation units is required.

The possible impacts resulting from the temporary implementation phase of the system must be evaluated both with respect to environmental aspects and in relation to the possible user discomfort deriving from construction site installation. The latter would negatively affect users both in terms of site fruition and the perception of the historical-cultural characteristics related to landscape quality.

Negative morphological and environmental impacts on the landscape produce indirect consequences on user perception, modifying users' well-being and leading to the abandonment of places with a qualifying landscape. Here, the social restriction (see Figure 3b) is intended as the control of users' well-being in the fruition of the place, and it is preserved through an impact evaluation aimed at analyzing new interventions in terms of service efficiency, user interface and symbolic perception of collective values. Symbolic-perceptive impact criteria are aimed at evaluating the relationship between the new intervention and the conservation of the existing signs, identity, and celebrative values assigned to a place by its local community, be it wide or not [17].

In many cases, user comfort is related to the total fruition of a service and its accessibility in terms of localization, timetable, availability, and positive feedback frequency. In fact, feedback encourages users to actively participate in its development processes. These aspects characterize the modalities and the user interface, which affect service efficiency in terms of times, costs, the complexity of resource flow circularization phases, and probability of unprogrammed service interruption caused by a short service life of the components of the system, and by a difficulty in substituting or repairing them. Acoustic impacts are certainly the most frequent ones, due to sound pressure and long reverberation. For example, due to the passage of compacting trucks between alleys of a dense urban context or during the loading and unloading phase of waste, in particular glass. Olfactory discomfort occurs in conditions of collection interruption and micro-climatic stress, leading to the emanation of harmful chemical-physical substances. In urban centers, air pollution is also caused by the high motor transport during resource collection, and by the lack of a systemic vision in the whole cycle management: in fact, product recycling and reuse would reduce this need.

Waste accumulation, related to an infrequent or undersized collection service, could lead to a health emergency. This process has frequently turned high-landscape-quality places into abandoned or decayed open-air waste deposits.

### 3.2. Subject Involved in the Waste Management Process

Population affects the landscape, while at the same time landscape affects the population as it produces emotions and feelings and stimulates the definition of meanings and values. In other words, it represents a key element in the quality of life of the population, in a reciprocal—or rather, circular—relationship. [18]. The vision of a circular city fosters the development of sustainable processes aimed at enriching landscape quality in its various dimensions: natural, anthropic, human, and social. The implementation of resource circularity requires synergy between public and private actors, with influences on the form and characteristics of the landscape. This synergy must be oriented toward a common purpose of sustainable development to produce strong effects on landscape quality: this benefits users' well-being.

The individuation of the stakeholders contributes to the indication of the strategies in circularization processes, in accordance with the hierarchy of interests and needs in the various groups of the community. Various studies have demonstrated the impact of stakeholders' role in the evolution of the circular process and have shown their importance-based interrelation [19–21].

In community strategies, enterprises and consumers are at the base of the transition toward a more circular economy managed by a local coordination system. In brief, two general categories of involved stakeholders can be defined; both are responsible, in different ways, for the development of the circularization of resource flows:

- economic promotors, that is entrepreneurs: these include investors, manufacturers, distributors, and retailers. These four sub-categories interact with each other both in the initial development process of goods from initial resources and with a final added value, and in the following recycling process of goods, through actions and services aimed at the economic optimization of their respective enterprise. Their mission can be associated with the development of market mechanisms to guarantee the minimum resource waste, the most efficient resource repartition, and, possibly, the correction of technological deficiencies with respect to innovation and strategic deficiencies with respect to creativity.
- users (consumers): these include the flows of residents and temporary inhabitants of a place. These two sub-categories have different needs, deriving from a common purpose: the fulfillment of individual well-being. To realize a sustainable future, guaranteeing the circular regeneration of resources without the consumption of new ones, this category must respect the strategies for: the reduction of waste production; reuse and recovery of a good or its parts; recycling and reuse, by differentiating product components.
- governance: is the institutional promotor and manages the articulation of the network of the involved subjects. Even more than being responsible, it must be able to activate circularity in the multiple aspects of a city. Specifically, its capability consists in: being able to foster the behavior of single subjects toward the realization of a shared idea of the common good; proposing operational strategies to the subjects, oriented at maximizing the potentialities of resources, negotiating their decisional autonomy; organizing relational systems between the subjects, minimizing their conflicts of interests and forming a conscious and participative consensus between actors.

In operational terms, the governance has a double role in coordinating enterprises: in addition to developing secondary resource markets and guaranteeing their correct functioning, it must create the conditions to allow entrepreneurs to exploit new potential markets related to the circular economy and ensure the availability of the required knowledge base on the job market. Concerning consumers, local administrations manage communication, information, and possibly incentivization processes, to address them toward an ecological path in the use of consumption of various products to perform conscious choices.

To limit the environmental footprint, the European Commission has invited administrations to form the Single Market for Green Products [22]. They have also described the methodologies to measure and communicate to the audience the environmental performances of products and processes. Economic promotors are invited to clearly define the potential environmental impacts of product life cycles, transparently communicating these data, and making them accessible to users. Governances must assess the quality of these data and provide minimum prescriptions and precise technical instructions to solve the criticalities in Life Circle Assessment studies. To the advantage of enterprises, green products can contribute to reducing production costs by using fewer resources; to the advantage of consumers, use and acquisition costs are also reduced.

### 3.3. Case Study: The Historical Center of Palermo, Sicily

The historical center of Palermo has a surface of about 250 hectares and is hence one of the largest ones in Europe; it has ancient origins, dating back to the Phoenician age.

The current state of the historical center is characterized by strong contrasts: its millennial stratification has led to a high morphological and spatial complexity in the urban layout. On one hand, this may produce notable difficulties in the introduction of systems and infrastructures; on the other hand, there is a high number of free spaces, which often

determines the fusion of public and private, as the latter still hosts a high concentration of public life.

The morphology of the urban fabric is constituted by a compact and alveolar urban fabric, with a dense and capillary system of roads and alleys characterized by difficulties in motorway traffic and, hence, complexity in resource flow collection management. In the '70s, the inhabitants started moving from the historical center to urban expansion areas: this led to a depopulation, which has been balanced in the last decade by an increasing presence of tourists and immigrated communities, and by a progressive diminution of productive activities.

The residual activities are related to the catholic diocese, located in the vast built religious heritage; the institutional functions of the Commune, mainly located in buildings overlooking squares and main streets; cultural and touristic functions, which have been the trigger for a slow recovery process of monumental building heritage; commercial activities, mainly represented by food businesses. Moreover, a gentrification process has taken place in limited regenerated areas of the territory; the evolution of this process could contribute to the reactivation of the historical center as a vital point of the city.

The peculiar morphological-structural nature of the city is shown by the strong visibility of the historical urban layout (irregular both in the articulation and in the size of the urban lots and streets) and the stratification of settlements. The area has limited natural features and no historical rural elements; instead, there are many historical buildings from the 17th century. Their building techniques are typical of the construction culture of Palermo at that time and their materials come from quarries that were once located in urban coastal areas. The landscape can be recognized as a structured system of interrelated elements, characterized by common linguistic-formal characteristics that make it highly sensitive to new possible transformations. Its historical center is a dynamic place, which houses intertwined public and private functions, with continuous trades of goods and services, and human flows; it is also a sharing place, an emotional center that mirrors the spirit of local culture.

Concerning MSW management, the study has been performed during the transition between two different collection systems: the historical center was served by a vehicular collection system, with localized residual waste dumpsters and waste banks for glass/metal multi-material, paper, plastic, etc. Wheeled bins were located near shops and were emptied daily with a micro-compactor. The local administration planned to introduce the curbside collection of paper and paperboard fractions, plastic envelopes, metallic envelopes, organic, and residual waste. The project has been implemented in December 2018. MSW has since been sent to the "Bellolampo" MSW Treatment Center, located around 30 km from the city center. The Center consists of a Mechanical Biological Treatment (MBT) plant, a leachate treatment plant, and two composting plants. As previously in the historical center, in several areas of Palermo waste sorting is still based on local posts of fixed and movable residual waste containers (dumpsters and bins). The long emptying times make these containers the main cause of environmental decay and landscape alteration, because of their notable physical footprint.

The collection is still performed by a truck on driveways and a mini-compactor on smaller roads: in collection and transportation phases, this leads to the release of high quantities of pollutants into the atmosphere, congesting the environment, which is by itself limited and poorly aerated.

The phase of production flow analysis, concerning frequency and quantity, is particularly important to plan collection, treatment, recycling, and disposal systems with respect to infrastructures, transportation, costs, times, and emissions into the atmosphere. There has been a progressive decrease in production in the City of Palermo, determined both by population decline and consumption reduction: currently, the average per capita value of 470 kg/inhab. per year. To perform a preliminary sizing of flows based on the data of total production per waste category in one year (year 2015), yearly and daily per capita quantities have been calculated, dividing the data by the number of inhabitants in the City

of Palermo and transposing them for the area of the historical center (HC in the Table) (see Table 3). The summation of the single production values for each category leads to an overall average per capita value of 1.36 kg/inhab · day.

**Table 3.** Synthesis of the per capita values per waste category in the historical center (HC).

| Recyclable Fraction | Total Production (kg/year) | Per Capita Production (kg/inhab · year) | Per Capita Production (Kg/Inhab · Day) |
|---|---|---|---|
| Paper | 4,463,390 | 6.388 | 0.018 |
| Paperboard | 1,569,550 | 2.246 | 0.006 |
| Organic | 10,167,560 | 14.553 | 0.040 |
| Mowing | 10,506,020 | 15.037 | 0.041 |
| Plastic | 2,577,120 | 3.689 | 0.010 |
| Metals | 150,720 | 0.216 | 0.001 |
| Batteries | 19,678 | 0.028 | 0.000 |
| Wood | 1,727,910 | 2.473 | 0.007 |
| Glass | 2,747,904 | 3.933 | 0.011 |
| Fabric | 19,400 | 0.028 | 0.000 |
| Toxic/flammable; HHW | 28,166 | 0.040 | 0.000 |
| WEEE | 2,877,211 | 4.118 | 0.011 |
| **Residual** | 310,037,890 | 443.752 | 1.216 |
| Population | 698,673 | | |
| HC Population | 28,682 | **Total kg/inhab · day** 1.360 | |

### 3.4. Urban Waste Management Scenarios

Waste sorting is the first phase of a circular process through which material flows are turned into resources, leading to a possible reduction of management costs [23] and environmental impacts, limiting landfilling and resource waste; this also reduces polluting emissions consequently. In an urban area, rethinking flow management can turn into an opportunity both from a physical-morphological standpoint, for the requalification of the area and its surrounding context, and from the social standpoint, involving users as an active part in the environmental regeneration process. These opportunities can be generated if the new planning keeps the identity of the urban landscape.

Flow management planning and infrastructural and technological choices can be improved by hypothesizing multiple alternatives. These alternatives must be realizable in the short term and must be suitable for executive planning on a given territory, to evaluate its real impacts and consequences. Intervention alternatives derive from an integrated design strategic process at various planning levels, connecting the strategic management of flow circularity and resource obtainment with the technological-operational dimension to be integrated into the historical urban landscape. In each hypothesized scenario, operational territorial planning tools and local development programming hence result to be complementary.

Impact evaluation has been performed on some scenario hypotheses, related to the phase of collection and resource flow pre-treatment. In particular, the individuation of a deeply different scenario seemed to be opportune, to highlight the differences between partial waste sorting (as in the historical center until December 2018, and currently adopted in some areas of the city) and the ideal state of continuous waste disposal. This latter has been developed according to the hypothesis that the flows are directly sent to the pre-treatment center, rapidly and with a tolerable extraneous fraction. Between the two opposed scenarios, one intermediate hypothesis is also considered: that is, curbside collection, which is also active in the historical center nowadays.

### 3.5. Scenario 1: Partial Waste Sorting

The first scenario is described through an analysis of partial waste sorting, which was present in the historical center of Palermo until December 2018 (see Figure 4a,b) and

is currently used in other parts of the city (see Figure 4c). In this collection modality, waste is mainly thrown in an undifferentiated way into mobile dumpsters; a small share is sorted (glass, plastic, and used clothes) in waste banks. The collection is performed with compacting vehicles that take waste to the MBT plant on the outskirts of the city.

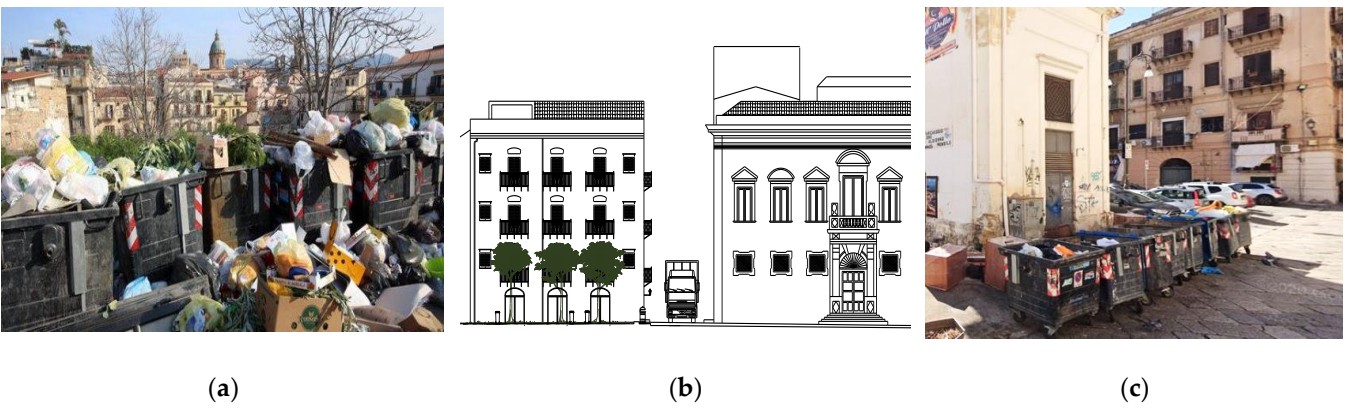

(**a**)　　　　　　　　　　　　　　　(**b**)　　　　　　　　　　　　　　　(**c**)

**Figure 4.** Partial waste sorting. (**a**) Dumpsters on the edge of the street in the historical center; 2015 image (Source: La Repubblica Palermo). (**b**) Graphical representation of the space occupied by a compactor truck in an internal street of the historical center (Source: Author's processing). (**c**) Current collection area in a square in the historical center of Palermo; 2022 image (Source: Google Maps).

The impact on the morphological characteristics of the landscape mainly derives from the visual burden of waste collection devices, as they require a high volumetric space that is often lacking in the historical center. The size of containers and the significant quantity of disposed waste represented a discontinuity in the relationships of the elements of urban space, as it interrupted the typological linearity of the built environment or hid some artistic details that characterize local history and culture. Moreover, there were evident interferences with viewpoints or panoramic routes, as some containers were located on squares or main roads; in some cases, their volume led to modify urban street sections, limiting their vehicle accessibility. These disadvantageous situations led to reflect on their high impact on the morphological dimension of the urban landscape.

According to 2018 estimates, this scenario resulted in a low percentage of sorted waste (10.48%) [24]. Another criticality was the greenhouse gas emissions from the huge quantities of street waste and the transportation means that collected disposed waste to take it to the plant. Polluting gas production sensibly altered the olfactory fruition of urban space, which is constituted by low-air-circulation areas: this led to a congested situation, with poor levels of atmospheric quality. The alteration of the environmental landscape had negative effects on acoustic comfort during waste collection and vehicle circulation, especially in streets with a reduced road section, where the resonance of compactor trucks could be perceived more. Despite having a temporary character over the day, these impacts occurred daily and negatively affect the environmental landscape quality, in addition to hindering the fruition of urban space or its parts.

The pedestrian movement was hindered at different times of the day and in different situations: during the loading phase of the waste from containers; during the circulation of these vehicles; if waste quantity exceeded the dimension of containers, it occupied part of pedestrian space on the sidewalks. This latter negatively affected user perception: when it occurred near accesses to buildings, it produced discomfort in terms of well-being and safety, as it hindered the full fruition of escape routes.

During the operation phase of the system, the outlined environmental and morphological aspects directly affected user perception, sensibly reducing the level of social well-being and the capacity of the place to recall expressive values associated with the city by the local community. The management phase did not include active user participation in the collection, as the system allowed a very limited differentiation of material flows. The loss

of symbolic values and celebrative image of the place, in addition to high user discomfort, led to a high social impact.

Environmental impact can be evaluated both qualitatively and quantitatively. The latter is possible by considering the value of greenhouse gas emissions per kg of disposed waste, that is 0.75 kg $CO_{2eq}$ [25], and then multiplying it by the estimated quantity, excluding the bio-stabilized percentage (10.48%): this leads to a total of 12,136,500 kg $CO_{2eq}$. The calculation of the polluting emissions is completed by adding those deduced from the expenses of transport on the routes from the containers to the treatment plant. The division of the total transportation cost by the unit price of fuel per liter, which is 1.4 €/L (2018 data), provides the amount of fuel consumed over one year, equal to 126,579 L. Finally, the multiplication between this value and the $CO_{2eq}$ emissions/diesel oil liter coefficient (2.65), leads to a total of 335,435 kg $CO_{2eq/year}$. Transportation costs have been determined by proportioning the quantity of disposed waste in the whole urban area, which is 302,426 t/year [26], to the quantities estimated in the analyzed area, equal to 22,746 t/year.

### 3.6. Scenario 2: Curbside Collection

The second scenario corresponds to the modality of waste sorting that is now active in the historical center. Curbside collection consists of the collection of the sorted waste categories of paper and paperboard, plastic envelopes, organic, and residual waste from the doorstep of households, and glass collection with bottle banks on the street. Compared to scenario 1, the sorted waste percentage has drastically increased, up to 63% [27], thanks to the involvement of citizens through a communication and information program that included home visits and the provision of documentation and tools required to achieve plan goals. However, the flows produced within an area with curbside collection cannot be entirely traced: this is due to a phenomenon named "migration" of waste to other areas that are not served by this collection system. This involves, in addition to an accumulation in other areas, that waste is hence not sorted in the correct manner, turning into residual waste. Curbside collection is among the most diffuse collection systems in the world; however, it is one of the most debated ones. The system efficiency and the incidence of overall circular management depend on several combined factors, among which is the quality of sorted material [28]. The model used in the historical center and other areas of Palermo is only partially resolutive, as it has led to a progressive increase in collected material quantity, resulting in undersized collection bins and the abandonment of residual waste on the street, in the space surrounding them (see Figure 5a–d). This is also due to a morphologically complex urban layout, where street dimensions do not allow locating additional waste containers and emptying them with bigger vehicles.

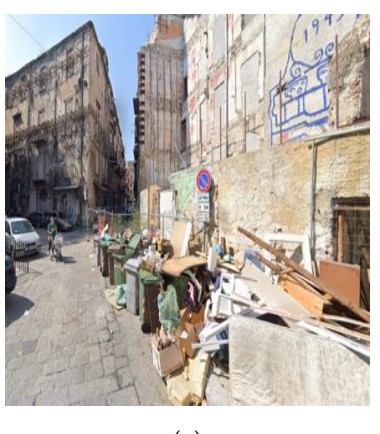 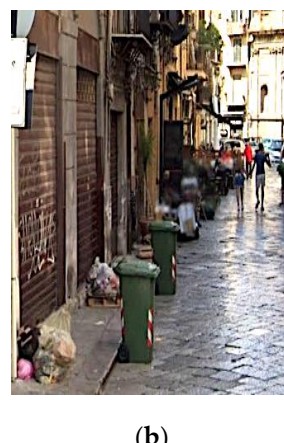 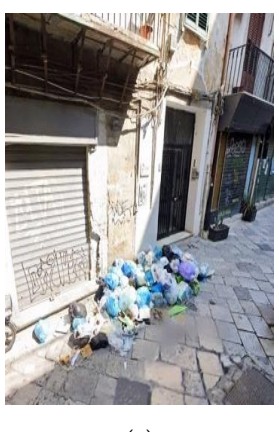 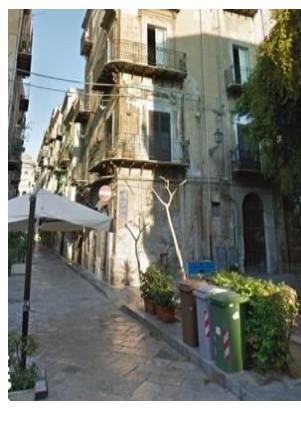

(**a**) (**b**) (**c**) (**d**)

**Figure 5.** Curbside collection in the historical center of Palermo. (**a**) Civic amenity site transformed into an open-air landfill. (**b**) Wheeled bins for organic waste, exceeding trash bags on the public land. (**c**) Deposits of condominium waste at the edge of a driveway. (**d**) The ideal state of the system, right after collection. (Source: Google Maps).

The impact on the morphological dimension of the urban landscape in scenario 2 is certainly lower than in scenario 1, but it is present. During the implementation of the system, there could be the need for the modifications of the internal volumes of the ground floors of the buildings where some containers are located. Hence, the linearity of street sections would be slightly affected by their infrequent collocation on the public land.

The vehicular collection has the same limits as for scenario 1; however, they are lowered by the possibility to use smaller vehicles. Specifically, it can alter historical-cultural relationships if the quantity of disposed waste exceeds the capacity of the containers and if they are in front of an urban space with strong typological connotations. Moreover, during waste exposure hours the number of containers is higher than in scenario 1: even though they have a smaller size, their chromatic characteristics lead to a visual focus. At some times of the day, curbside collection negatively affects the conservation of the readability and recognizability of the landscape quality of the urban system, leading to a medium impact on the morphological dimension of the landscape.

From an environmental standpoint, the percentage of the sorted waste is considerably higher than in scenario 1 and this could allow a strong reduction in polluting emissions. However, there are still vehicular emissions during the phase of collection and transportation to the treatment plant, and emissions by the fermentation of the collected material when collection times are delayed. Their main consequences are an alteration of olfactory sense and atmospheric pollution, increased even more by the higher number of collection posts, even though the quantity of sorted waste in each of them is lower than in scenario 1. Due to these limits, the service might alter environmental levels, hindering the full fruition of part of the urban space.

Curbside collection enhances participation in the collection if the system is combined with an information and communication campaign addressed to citizens. In this way, they can be fostered toward an increasingly more efficient performance of the system, as they consider themselves an active part in the sustainable regeneration process of their living place. Even though the engagement in the collection represents an excellent advantage in the interface between the system and its users, its implementation has required significant initial investment for the acquisition of vehicles and containers. Further costs could be caused by the wear and tear of containers and by the long times for resource recovery, as the containers are scattered in the urban system.

In environmental terms, the quantitative evaluation is coherent with the qualitative analysis, as the greenhouse gas emission values result to be lower than in scenario 1 over the management cycle, yet not irrelevant. Considering the quantity of collected material al net of the percentage of recycled waste or that subjected to energy recovery, emissions are equal to 3678,750 kg $CO_{2eq}$.

The quantity of fuel (in liters) consumed by the vehicles for the collection and the transportation to the treatment plant has been calculated from the costs; then, after applying the $CO_{2eq/l}$ coefficient, the result is 453,685 kg $CO_{2eq/year}$.

### 3.7. Scenario 3: Automated Vacuum Waste Collection System

This scenario is based on the hypothesis of an automated vacuum waste collection system, patented by the Envac group (https://www.envacgroup.com/ accessed on 8 May 2022) and already active for years in various cities of the world and also in historic centers such as Vitoria-Gasteiz, Seville, Barcelona (Spain), Stockholm (Sweden), Bergen (Norwey), etc. The scenario is sized with the same percentages estimated for curbside collection. The design hypothesis is an automated vacuum collection network in the historical area of "Mandamento Castellammare", overlooking the port area of historic center of Palermo. This district has a more regular urban layout dating back to the 19th century and is in the border area between the city and more morphologically complex areas from the Middle Ages.

The system uses air pressure difference to transport waste in underground pipelines from their original location to central processing facilities, placed at strategic points for the sorting of urban flows. Three collection stations have been hypothesized for this

area, considering about 25,000 equivalent apartments. An apartment is intended as a house with a surface of 100 m$^2$ destined for the performance of one or more functions. The quantity of material production in the area has been calculated by multiplying the number of apartments (applying a coefficient related to the in-use destination) by the average production quantity estimated for each apartment. This latter corresponds to the per capita daily production, multiplied by the average number of inhabitants in the reference apartment, which is 2.17. The multiplicative coefficient provides a clear result in the areas where the presence of specific functions (for example, food businesses) decisively affects production. The waste categories can be collected through a continuous system by various waste intake hatches: in the current hypothesis, there are four (organic, plastic, paper, mixed) for each collection point. Waste is directly transported through a single general network below the street level, without any preliminary pre-treatment, which is instead performed at the arrival at one of the collection stations. The system does not require operational personnel, as it is managed and controlled through software linked to a web platform that allows remote control. This software operates progressively on the branches of the network, after the activation of a sensor that indicates the achievement of the maximum capacity in disposal containers [29].

For each branch, the software inspects all the elements of the system before starting the collection process, verifying the absence of problems and anomalies. Then, it regulates the opening and closure of a valve located at the intersection between a vertical pipe in the container and a horizontal one for transportation and emptying. After valves open, the automated system aspires waste from the initial point to a terminal collection station located in a strategic area of the district, with favorable characteristics for waste sorting. The system mainly consists of: three central processing facilities; the waste transportation network; thirty waste collection points. All the components required for the collection process and its supervision are installed in the processing facilities where all the flows converge through the transport by automated vacuum collection. The network connects the processing facilities with the collection points. The collection points consist of portholes (see Figure 6a,b) for the collection of waste categories and a vertical pipe that connects the inlet valve with the main pipeline. The pipelines are organized in a modular grid that can be adapted to the morphology of the territory to realize an underground pipeline to converge waste into hermetically closed containers, before being transported to the final treatment centers. Transportation networks can be public or private depending on the property of the area where they are installed; they are generally located 2.50 m below the street level and their laying surface can be the street subsoil or parts of the buildings, such as garages, crawl spaces, etc. The pipeline connects collection points with the processing facility through a closed-loop network that follows building blocks.

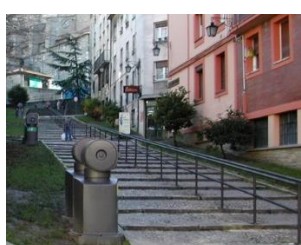

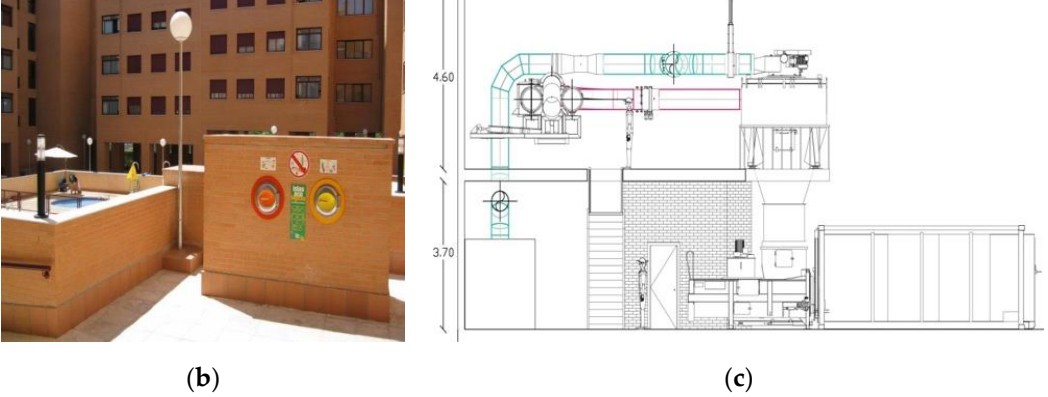

(**a**)  (**b**)  (**c**)

**Figure 6.** Automated vacuum waste collection system. (**a**) Columns for waste disposal in the urban center of Vitoria-Gasteiz, Spain. (**b**) Columns for waste disposal integrated into buildings, Madrid, Spain. (**c**) Waste separation process at the arrival at the collection station (Source: Envac Iberia S.A.).

The waste drops into containers from the pipeline by gravity and is then compacted; full containers are moved with horizontal and vertical translations by a bridge crane that substitutes them with empty ones. The last operation is the transportation of full containers to recycling and/or landfilling centers by trucks. There, they are emptied again and brought back to the collection station (see Figure 6c). The system is on cooldown between the collection process of one branch and another; however, it is available to users 24 h a day and they can throw their waste continuously, whenever it is more convenient to them.

The parameters for the sizing of the system are the following: the current disposal peaks over the day and the expected increase of the disposed waste in the future, considering specific territorial needs (e.g., tourism or overcrowding at specific times). In each project, two typologies of collection processes can be used: a fixed-time collection process set according to the waste disposal peak hour, and a fixed-level collection process for unexpected material quantities. When the system is set on the maximum level of capacity of the valves and one valve indicates the achievement of this threshold, the system does not only collect waste from full valves, but also the whole waste of the served area, to make energy consumption more efficient.

Even though the pipelines follow a modular grid, which can be adapted to the conformation of the territory and realized through partial installations, there is a strong morphological impact during the implementation phase of the system, represented by its underground infrastructure. This could produce an alteration of the historical sediments—when present—during the phases of removal and restoration of the pavement, also for preventive diagnostic investigations. That would lead to the unexpected interruption of operation, and to the need of planning alternative solutions. During system operation, disposal is performed through small-sized inlets, whose design can be varied according to the characteristics of the installation areas. In this situation, the morphological impact on the urban space is minimal, aside from the modifications of the internal volumes of buildings chosen as collection stations. However, these could be turned into opportunities for the restoration of decayed buildings, safeguarding and valorizing the historical-cultural qualities within them and the surrounding urban landscape. This might lead to conclude that the impact on the morphological dimension of the urban landscape is low.

Regarding environmental impact, there is a drastic difference between the construction and the operation phase. Concerning the former, acoustic pollution monitoring is required during the construction and maintenance phases; concerning the latter, it already complies with the principles of environmental sustainability, as it excludes polluting emissions in waste collection and transportation in the historical center. In the collection station, resource flows are separated from the airflow through separation cyclones: air is aspirated in a filter room, where dust is removed with a physical filter; then, it is purified from smells with an activated carbon filter and is inflowed into a silencer before being outflowed back in the environment, to limit the acoustic pollution of the system. Greenhouse gas emissions, in terms of waste production quantity, are the same as in scenario 2, corresponding to 3678,570 kg $CO_{2eq}$. There is a strong reduction in emissions during the collection and transport phases, as fuel consumption is limited to the route from the collection stations to the treatment plant. In particular, the total cost for the transportation from the three stations is equal to 5026 €, corresponding to 9513.5 kg $CO_{2eq/year}$.

Considering the same goals of sorted waste percentage as for curbside collection and, likewise, a good user sensibilization campaign, a high level of participation can be expected, also thanks to the possibility of continuous disposal (without time slots and quantity limits). However, there are disadvantages concerning the investment cost and the complexity in the repair and substitution of components, due to the presence of special pieces under the roadway. These limits are compensated by a low failure probability and by the potential increase in incomes, thanks to the velocity of obtainment of pre-treated material flow. In general, social impact is very low, aside from interferences with the perception of the place during construction and/or maintenance phases.

## 4. Results

The results (see Figure 7a–e) derive from an analysis performed on the criteria referred to the morphological, environmental, and social dimensions of the urban landscape. After analyzing the scenarios one by one, they are compared through pairwise comparisons with the Analytic Hierarchy Process (AHP). The choice of this method fits the multi-criteria nature of the study, in which the urban landscape is an organic system constituted by multiple elements.

The hierarchical organization of the elements that concur in the fulfillment of the main goal results to be useful to manage the evaluation according to criteria with a limited entity, and a direct influence on the landscape in their specific dimension.

The three scenarios are compared through pairwise comparisons for each of the criteria, to determine which one determines a higher impact on the urban landscape through its management and infrastructural configuration, for each of the dimensions. The results of the comparison are the dominance coefficients $a_{ij}$, which represent the quantitative values (weights) to graduate the preferability of a scenario $i$ to another scenario $j$.

The weights of the scenarios for each criterion (local weights) are calculated with the eigenvalue power method for the main eigenvector. The result is a single vector for each scenario and each criterion. The summation of the components of the vectors provides a single vector; its normalization produces a unitary versor. The versor allows obtaining the preferability rate of a given scenario for each criterion: $w_1/w$; $w_2/w$; $w_3/w$. Local weights are multiplied by the weights of the corresponding superordinated criteria, hence obtaining the values of the global weights. In this research work, the weights of the criteria depend on the importance of each criterion for the achievement of the general goal. This importance has been determined through a pairwise comparison between the criteria during the hierarchical organization phase of the analysis model (see Figure 8a–d).

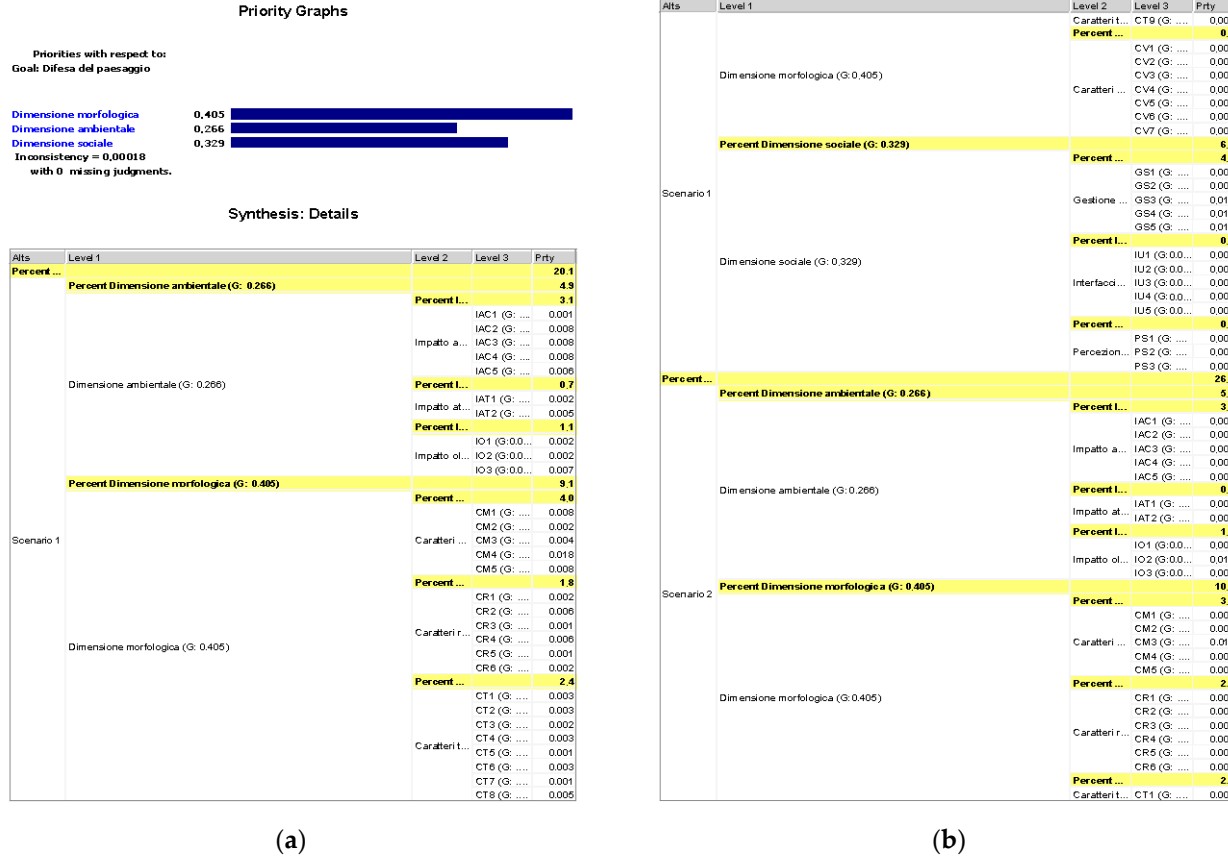

(**a**)　　　　　　　　　　　　　　　　　　(**b**)

**Figure 7.** *Cont.*

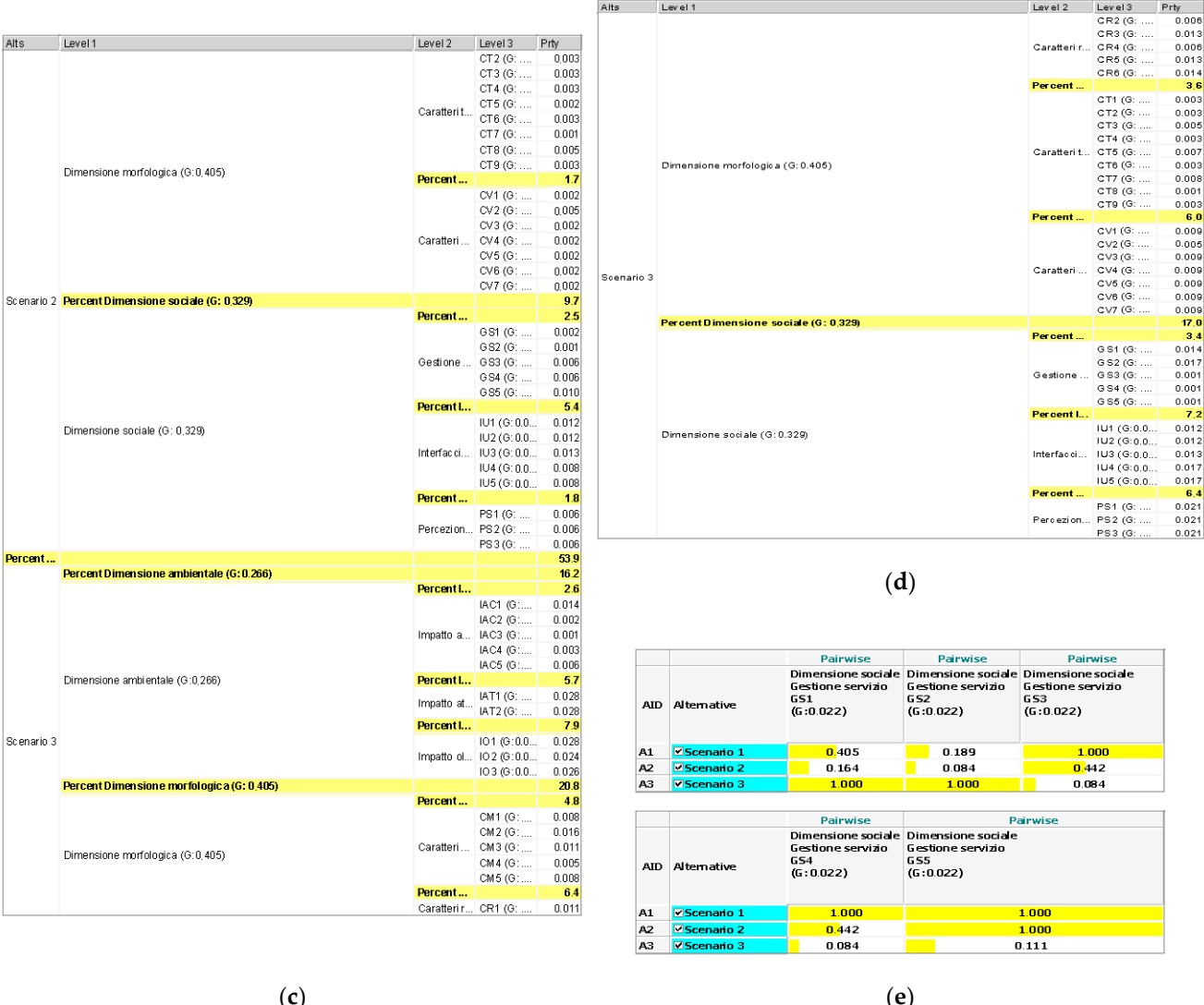

(**c**)　　　　　　　　　　　　　　　　　　　　(**e**)

**Figure 7.** Synthesis of results: goal fulfillment rate for each scenario, dimension, and criterion. (**a**) Data related to scenario 1, with an overall 20%. (**b**) Data related to scenario 2, with an overall 27%. (**c**,**d**) Data related to scenario 3, with an overall 53%. (**e**) Extract of the results of the pairwise comparison: consider criterion GS5 (service implementation), where the goal is completely fulfilled in scenarios 1 and 2 (with a simple implementation, as they do not require complex technological infrastructures) and is associated with a value of 1; compared to the other two, scenario 3 is more complex as it includes an articulate network below the roadway and so its score is lower than the other two by 90%. In figures (**b**,**c**), respectively, scenarios 1 and 2 have a value of 0.1 for the criterion GS5, while in figure (**d**) an approximate value of 0.01 is visible for scenario 3. The criteria are expressed in Figures 1 and 2. (Source: Author's processing). For (**a**–**d**), the name of the indicator relating to the analyzed dimension is abbreviated in the Level 2 column (the full name is visible in Figures 2 and 3). In the Level 3 column the indicator is identified with a code.

In this evaluation, it has been hypothesized that both the level of the three dimensions individuated for the general goal and the level of the criteria have the same influence on the achievement of the general goal. The pairwise comparisons have shown that, compared to scenario 1, both curbside collection and automated vacuum collection systems have higher advantages for the protection of the urban landscape. The most impactful gap is related to the environmental dimension, for both scenarios (see Figure 9a).

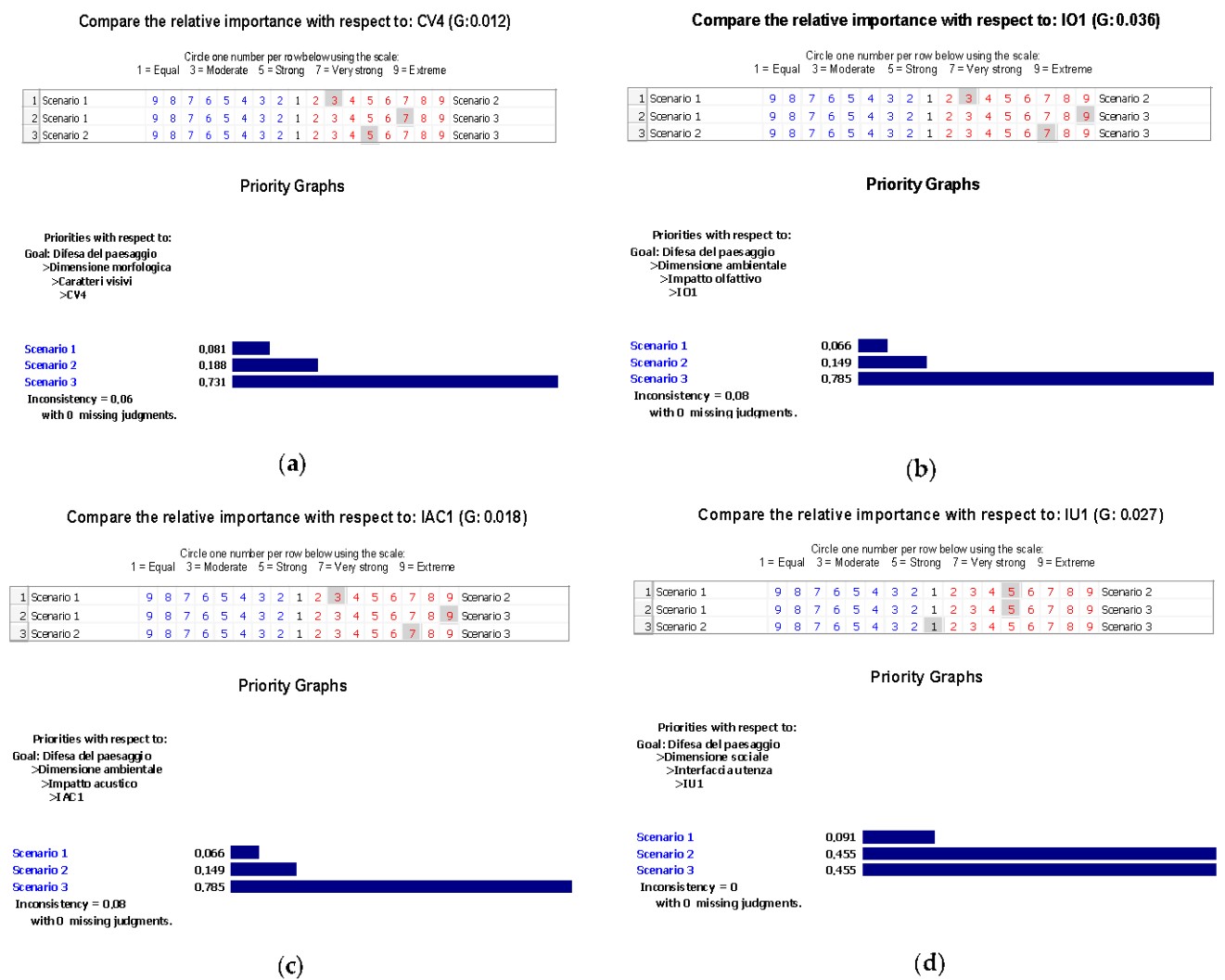

**Figure 8.** Extract of the pairwise comparisons between the scenarios. (**a**) Reduction in panoramic view. In scenario 1, the reduction is lower than in scenario 2 but much higher than in scenario 3; in scenario 2, the overall perception is strongly limited, compared to scenario 3. (**b**) Exposure of fermented waste to the external environment. In scenario 3, the collected waste is never exposed to the external environment as the automated vacuum collection system can receive high waste quantities. Instead, this regularly happens in scenarios 1 and 2. In scenario 2, this might happen less extensively as the users of each collection device are estimated to be less than in scenario 1. (**c**) Negative variation of the average daily number of vehicles. During the collection phase, scenario 3 does not require vehicular traffic as the system works through automated vacuum waste collection in underground pipelines. Hence, the achievement of the overall goal is higher than in scenario 2 and much higher than in scenario 1. (**d**) Limited user accessibility. In scenarios 2 and 3, the degree of accessibility is the same, as in both there is the possibility to include waste delivery points in housing complexes. The difference with scenario 1 is the same for both scenarios 2 and 3. (Source: Author's processing).

Concerning the morphological dimension (see Figure 9b), the second scenario has visual disadvantages during the exposition of the containers, unless they are placed in a hidden post and if the disposed waste quantity does not exceed the limit capacity of the containers. The third scenario can be considered uninfluential during the operation phase, as its waste delivery points can be dissimulated among landscape elements and provide an indifferent contribution to the characterization of the urban landscape. As outlined, this is not true during the implementation phase of the system, which presents possible alterations in perceptive and environmental terms. Both scenarios 1 and 2 can affect the relational

continuity of the historical-cultural characteristics of the landscape, slightly differing by the duration of the exposure of collection devices. In general, the automated vacuum collection scenario is advantageous; however, a focus must be given to the system operation, through diffuse on-site diagnostic campaigns.

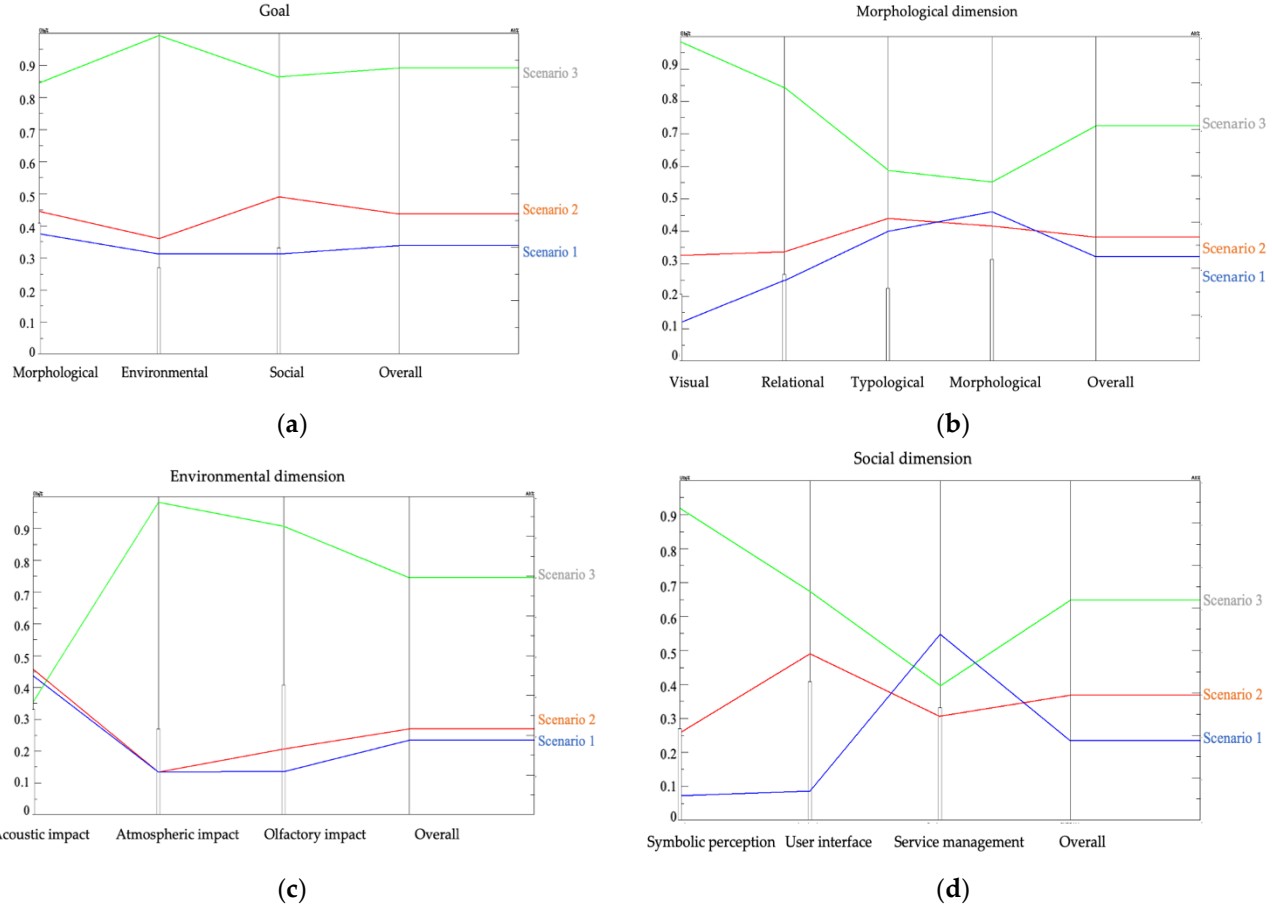

**Figure 9.** Graphical synthesis of the results by dimension and scenario. (**a**) Morphological dimension. Various criteria are related to the conservation of the characteristics of the urban landscape. For example, in scenario 3 the installation of the pipelines implies a modification of the roadway, with a possible alteration of the pavement. In scenarios 1 and 2, during waste delivery hours there could be influences on the readability of the typological and constructional characteristics of the building, up to the distortion of unique elements that characterize the historical and/or cultural image. (**b**) Environmental dimension. One of the relevant factors is the acoustic impact, considered in the evaluation criteria with respect to the implementation, operation, and—if present—maintenance phases. The composition of positive and negative aspects for each criterion reveals that the scenarios are almost equivalent. (**c**) Social dimension. The linear anomaly point in the diagram can be related to the difficulty in implementing the service and to the additional difficulty in the obtainment and substitution of components. (**d**) General goal. Scenario 3 is closer to the general goal, especially in relation to the environmental sphere of the urban landscape. (Source: Author's processing).

Maintenance works would lead to the installation of a temporary construction site in the area and the potential use of condominium ground floors, modifying their internal volumes. While this last condition is only possible for scenario 3, it is necessary for the realization of scenario 2, which requires individuating suitable spaces for the collocation of collection devices that are not on the street. The service consists of reversible elements in scenarios 1 and 2 and it is located underground in scenario 3. So, it never leads to the alteration of material or typological characteristics, but rather to a disturbance. The latter can be limited with actions of chromatic and volumetric integration of collection devices,

precautions through spatial planning, and adequate management design in terms of service dimensioning, in relation to the entity of collected waste.

The environmental evaluation (see Figure 9c) has reported a huge gap between scenario 1, with partial waste sorting, and the other two, with total waste sorting, with reductions of around 70% in terms of greenhouse gas emissions from collected waste. The automated vacuum collection scenario shows a decrease in emissions by 98%: this is a clear environmental benefit with respect to yearly transport consumption, as vehicles only circulate outside the historical center to take pre-treated material to recycling plants. Concerning transport, curbside collection would lead to a disadvantageous framework, because of the increase in the number of waste delivery points; however, it would be compensated by the benefit related to the higher quantity of sorted waste than in scenario 1.

The evaluation also considers the factors related to the typological composition of sorted waste. Such factors could lead to the insurgence of fermentation phenomena, with the consequent generation of malodorous, polluting gases. The acoustic impacts are analyzed in two different phases: system implementation and operation (see Figure 9d).

In both phases, the performance of scenario 3 is quite contrasting, compared to scenarios 1 and 2: while in the initial phase scenario has a notable environmental impact and requires sound level monitoring, then sound emissions are completely zeroed. In conclusion, these conditions lead to even acoustic impacts in the three scenarios; hence, the definition of a choice would require a user perception-based evaluation. Another advantageous perspective, compared to scenario 1, lies in interface and user perception: a waste sorting service with integrated systems would involve citizens in the regeneration of their living place. The analysis of the service management highlights potential difficulties with respect to maintenance in scenario 3; even if the expected service life of the components of the system is equal to 10 years, they would be difficultly obtainable because of the presence of numerous special components. Moreover, interventions would have to be performed below the roadway, requiring construction site installation, and securing of the area (see Figure 8c).

## 5. Conclusions

The multi-criteria model applied for the evaluation impact of the hypothesized scenarios on the urban landscape is based on the identification of multiple subsystems and characteristic elements within a landscape. The urban landscape has been analyzed as a system, whose elements are part of a complex network of physical and socio-economical relationships. Landscape components are inter-dependent, reciprocally influential, and have circular relationships; each component is evenly important and necessary [30]. Just think of the environmental parameters (greenhouse gases, $CO_2$ concentrations, and sound emissions) that affect the physical well-being of the resident and temporary communities of a place, in addition to the conservation of the urban space.

A sustainable landscape must be resilient to new transformations and the applied method evaluates this capacity by analyzing urban balance and the vulnerability of its components in three different situations: impacts on morphological relationships, environmental impacts, and social satisfaction measurement.

Scenario analysis has shown the disadvantages of scenario 1, compared to the various advantages produced by the alternative solutions with respect to multiple evaluation criteria, in addition to their planning feasibility. This method stands as a support to specialized professionals with respect to operational difficulty in an urban landscape, envisaging the consequences produced by the implementation and operation of the scenarios.

The benefits from recycling or energy recovery and the reduction of vehicular distances during waste transport have been estimated in environmental terms. The same percentage goals and sorted quantities have been chosen for the second and third scenarios. The analysis has allowed operating along an analytical process by graduating incidences and generating quantitative judgments from a qualitative logic. The method is repeatable in

other contexts: it is a decision support tool, to be implemented with specific evaluation criteria for the area of intervention and the typology of hypothesized scenario.

Resource flow management is mainly a social issue and can be solved best if it is interpreted as an opportunity for social participation and development goals sharing. In environmental terms, the goal is the sensible reduction in $CO_2$ emission, by increasing recycling through a virtuous waste sorting, producing renewable energy from biomasses, and substituting fossil energy with it. In economic terms, it can be demonstrated that recycling strongly influences management costs and can be turned into an incentive to increase waste sorting. In urban planning, the new technological systems represent an opportunity for spatial regeneration, by valorizing an reality whose relationships are preserved over time.

The multicriteria evaluation approach adopted in this paper can be replicated at various planning levels and for various service typologies: resource flow management service is just one of them. The key aspect is the method: that is, the approach to the conservation of the cultural capital of a city, keeping in mind all its characteristic aspects that could be altered. The cultural capital of a city indirectly contributes to the stability and resilience of an urban eco-system: as such, it has an intrinsic value related to its production of social capital [31]. The social capital is the glue that keeps together the various subjects, as a reflection of the collective accumulation of knowledge, creativity, and values recognized by the community in its living places.

**Funding:** This research was funded by the Italian Ministry for the University with the project PON "Research and Innovation 2014–2020" Section 1 "Researchers Mobility" with D.D. 407 of 27 February 2018 co-financed by the European Social Fund—CUP B74I19000650001—id project AIM 1890405-3, area: "Technologies for the Environments of Life".

**Institutional Review Board Statement:** Not applicable.

**Informed Consent Statement:** Not applicable.

**Conflicts of Interest:** The authors declare no conflict of interest.

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
