# Peer review of "The Circularity of MSW in Urban Landscapes: An Evaluation Method for a Sustainable System Implementation"

_sustainability, doi:10.3390/su14127358_

Round 1

Reviewer 1 Report

The author presents a useful framework for addressing enhanced waste/resource cycling in dense cities. Overall, the evaluative method presented is logical and thorough, though perhaps too skimpy on human health impacts as compared to visual/spatial impacts on local environs. The text is quite dense in some sections, and at times the jargon is tiresome.

I've provided detailed commentary in the attached.

Overall, I enjoyed reading the manuscript and thank the author for their contribution to the journal!

Author Response

Dear Reviewer,
thank you very much for your valuable comments, I have tried to correct the text in almost all points but some I have had difficulties. For example the enlargement of the figures is not advisable as they have a not excellent resolution, in any case I have enlarged them slightly; the impacts on human health are considered transversally in all dimensions (olfactory impact, air quality, etc.) which naturally affect various aspect of human health; figure 7 and 8 unfortunately they are tables extracted from the software that I cannot modify and translate.

Reviewer 2 Report

Dear Elvira

I congratulate you on your manuscript. I will recommend it to the editor that it be published, and there are some changes I suggest to you, below:

1) The title is too generic, and does not mention municipal solid waste. A new headline, newspaper headline style, spelling out very clearly the most important message of your article would be better;

2) Please include more keywords in MS. This increases the visibility of the article in search engines, and makes it easier for readers to understand the topics covered;

3) I recommend that you include a paragraph in the Introduction to explain what cities are, from an ecological point of view. I mean, cities are heterotrophic ecosystems, with huge input and output environments (i.e., with huge ecological footprints). Therefore, reducing their ecological footprints is essential for urban sustainability. The inclusion of this paragraph is a robust justification for your research;

4) Figures 2, 3, 4, 5, 6, 7 and 8 are poorly legible. Why are figures 4, 5 and 6 so small?

5) The inclusion of an infographic that translates the goals described between lines 75-93 would increase readers' understanding of the objectives of your study;

6) Please avoid acronyms like LCA (line 330). They make it difficult to read and understand the text;

7) The inclusion of an infographic or photos of the Historic Center of Palermo would make the text easier to read and understand;

8) On line 391 (and Table 2): 470 kg/pax per year. What is pax?

9) Line 553: “Mandamento Castellammare”: is this location in Palermo? Your wording is not clear on this;

10) It is unclear - at least to me - why CO2 emissions are lower in the Automated vacuum waste collection system (lines 642-645). Do they pack more waste, and thus require less trucking?

11) Automated vacuum waste collection systems seem like a fantastic solution to the ubiquitous urban solid waste generation dilemma. However, you refer vaguely to the economic costs of installing those infrastructures. I suppose those systems are very expensive, aren't they? In lines 775-777 you state that "The urban landscape has been analyzed as a system, whose elements are part of a complex network of physical and socio-economical relationships. Landscape components are inter-dependent, reciprocally influential, and have circular relationships; each component is evenly important and necessary", but I haven't found a more detailed comment on how much an Automated vacuum waste collection system would cost. Who would finance it, the City of Palermo?

12) Maybe that automatic system is economically viable in Europe, but since Sustainability is an international journal, I recommend that you include a paragraph in the Discussion, about its adoption in cities of developing countries with a huge generation of waste. Brazil, for example.

All the best,

Fabio Angeoletto (UFMT, Brazil)

Author Response

Dear Reviewer, Thank you very much for your important advices. Where it was possible I gladly made the changes. Unfortunately I would not like to add other paragraphs as according to another reviewer it is already very long as a contribution. Consider that it is an excerpt from a very extensive research work where other case studies, including international ones, have also been considered, but I must somehow restrict the text of the paper. I zoomed in on the figures but they don't have excellent resolution. The costs of all three systems were also analyzed during the research but I chose not to report them in this contribution because they derive from a long and detailed metric calculation that would require an equally long separate paragraph.
I wish you the best too.

Reviewer 3 Report

The paper consists serious work, well-defined methods, serious analyses and results corresponding with expectations of the author. There is no  too much to comment details in this respect.

What is to be mention as serious conceptual comment and question concerning the whole work is the following:

1.  The method is described by quite complex system-oriented way, using formal procedures, implementing really a number of criteria described on Fig.1, 2, 3 (by the way the Figures are hardly readable). It would be praiseworthy, but ...  To follow correctly the flow of the formal process the criteria had to be indexed, and, to be formally identified in the verbal description of the scenarios of the case study in Palermo. It is not the case, so the description of the real situation in Palermo looks like an ordinary evaluation of the problems of waste management wherever else. The relation of the method and criteria to the scenarios is difficult to disclose.

2. There is the attempt to connect the above-mentioned gap in the chapter of results in form of Figure 7 and 8. There are listed the indexes of criteria, but not explained, neither explained the process show the results had been gained. More-over the tables on the figures contents abbreviations in Italian language. Providing that the results be based on correct calculations, this part must be consistently improved.

3. In spite that I appreciate very much the system-oriented approach to this question, even being susceptible of using qualitative and semi-quantitative characteristics for quantitatively shaped process and formulas, I have to put a principal question. Had the author before starting the formal evaluation any doubt that the scenario 1 is the worst and the scenario 3 is the best in overall respect? By other words: are so heavy “weapons“ needed for this result?

There would be a good reason: the case, when the relative differences between scenarios would be presented by economic/financial values.

General conclusion: I recommend principal improvement and clear explanation of the relations between analyses, criteria, description of scenarios, discussion and results.

Author Response

Dear Reviewer, Thank you very much for your valuable advice. Before starting the research I could have assumed that scenario 1 would be the worst but I had a strong doubt that 3 was better than 2 due to the difficult application in the historic center. The work you see is an excerpt from a very long research that compared the scenarios also from an economic point of view, also calculating the payback time. I chose not to deal with the economic part in this contribution as it is very long and somehow I had to make a choice on the aspects in which I intended to focus in relation to the special issue of the review. I have tried to clarify the method and the choice of criteria.  If you see also in figure 9 the method of comparison between the criteria is explained. Only one example is given but the analysis procedure is the same for all criteria.
Thanks again and best regards.

Round 2

Reviewer 3 Report

The paper has been improved.